# Causal effect of atrial fibrillation/flutter on chronic kidney disease: A bidirectional two-sample Mendelian randomization study

**Masahiro Yoshikawa**[1]*, **Kensuke Asaba**[2], **Tomohiro Nakayama**[1]

1 Division of Laboratory Medicine, Department of Pathology and Microbiology, Nihon University School of Medicine, Tokyo, Japan, 2 Department of Computational Diagnostic Radiology and Preventive Medicine, The University of Tokyo Hospital, Tokyo, Japan

* myosh-tky@umin.ac.jp

**Data Availability Statement:** All data are available from: https://ckdgen.imbi.uni-freiburg.de/ and https://gwas.mrcieu.ac.uk/.

## Abstract

Chronic kidney disease (CKD) and atrial fibrillation are both major burdens on the health care system worldwide. Several observational studies have reported clinical associations between CKD and atrial fibrillation; however, causal relationships between these conditions remain to be elucidated due to possible bias by confounders and reverse causations. Here, we conducted bidirectional two-sample Mendelian randomization analyses using publicly available summary statistics of genome-wide association studies (the CKDGen consortium and the UK Biobank) to investigate causal associations between CKD and atrial fibrillation/flutter in the European population. Our study suggested a causal effect of the risk of atrial fibrillation/flutter on the decrease in serum creatinine-based estimated glomerular filtration rate (eGFR) and revealed a causal effect of the risk of atrial fibrillation/flutter on the risk of CKD (odds ratio, 9.39 per doubling odds ratio of atrial fibrillation/flutter; 95% coefficient interval, 2.39–37.0; $P = 0.001$), while the causal effect of the decrease in eGFR on the risk of atrial fibrillation/flutter was unlikely. However, careful interpretation and further studies are warranted, as the underlying mechanisms remain unknown. Further, our sample size was relatively small and selection bias was possible.

## Introduction

Chronic kidney disease (CKD) is a major global burden, and 1.5% of the total deaths worldwide were attributed to CKD in 2012 according to the World Health Organization [1]. CKD is principally caused by diabetes, hypertension, and glomerulonephritis. The comorbidities of CKD include anemia, bone disease, cancer, and cardiovascular diseases [1]. Atrial fibrillation (AF) is the most common arrhythmia and is a major burden on the health care system worldwide, as this condition can cause ischemic stroke and cardiac dysfunction [2]. Several observational studies have reported clinical associations between CKD and AF [3–8]. However, the causal relationship between CKD and AF remains to be elucidated as traditional observational studies lacking randomization designs are typically prone to bias due to various factors including confounders and reverse causations [9].

**Funding:** The authors received no specific funding for this work.

**Competing interests:** The authors have declared that no competing interests exist.

Mendelian randomization (MR) is an epidemiological method that mimics the design of randomized controlled studies using single-nucleotide polymorphisms (SNPs) as instrumental variables (IVs) and is used to examine the causal effects of a risk factor on the outcome of interest. As genetic variants such as SNPs are randomly assigned at conception according to Mendel's law, MR studies are not influenced by confounders or reverse causations and can overcome the limitations of observational studies [9]. In this study, we conducted bidirectional two-sample MR analyses using publicly available summary statistics of genome-wide association studies (GWASs) to investigate causal associations between the risk of atrial fibrillation/flutter (AF/F) and the change in serum creatinine-based estimated glomerular filtration rate (eGFR) or the risk of CKD for the first time in the European population.

## Methods

### Study design and data sources

We performed bidirectional two-sample MR analyses that included (1) an MR analysis estimating the causal effect of the risk of AF/F (binary data) on the change in eGFR (continuous data), (2) an MR analysis estimating the causal effect of the risk of AF/F (binary data) on the risk of CKD (binary data), and (3) an MR analysis estimating the reverse causal effect of the change in eGFR (continuous data) on the risk of AF/F (binary data). All analyses were conducted using the TwoSampleMR package (version 0.5.5) in R software (version 4.0.3) [10]. A *P*-value below 0.017 (0.05/3 by Bonferroni correction) was considered statistically significant and a *P*-value between 0.017 and 0.05 was considered suggestively significant in the three MR analyses. A *P*-value below 0.05 was considered statistically significant in the MR-PRESSO (Mendelian Randomization Pleiotropy RESidual Sum and Outlier) global test and outlier test.

For eGFR and CKD datasets, summary statistics were available from the GWAS meta-analysis performed by the CKD Genetics (CKDGen) consortium [11]. The dataset for eGFR used continuous data of log (eGFR) and included 567,460 participants of European ancestry [11]. Serum creatinine assays were described in the GWAS study [11]. GFR was estimated using the Chronic Kidney Disease Epidemiology Collaboration equation on adults (> 18 years of age) and the Schwartz formula on individuals who were 18 years old or younger, respectively [11]. The dataset of CKD (defined as eGFR < 60 ml/min/1.73m$^2$) used binary data of log odds ratio (OR) and included 41,395 cases and 439,303 controls of European ancestry [11]. These two datasets are publicly available from the "Wuttke et al. 2019 publication files" uploaded in the CKDGen Meta-Analysis Data as the file names "20171017_MW_eGFR_overall_EA_nstud42. dbgap" and "CKD_overall_EA_JW_20180223_nstud23.dbgap", respectively [12].

For the AF dataset, summary statistics were available from the largest GWAS meta-analysis published by Nielsen et al. that included 60,620 AF cases and 970,216 controls of European ancestry [13]. However, Nielsen's GWAS meta-analysis included the deCODE study with 13,471 AF cases and 358,161 controls. In contrast, the CKDGen GWAS meta-analysis also included the deCODE study with 15,939 CKD cases and 192,362 controls [11]. If we use Nielsen's GWAS meta-analysis, at most 22.2% (13,471 out of 60,620) of the AF cases may overlap with participants in the CKDGen GWAS meta-analysis. Sample overlap in cases between the exposure and outcome datasets can lead to substantial bias in the causal estimate of MR studies in the direction of both the null and the observational association [14]. Therefore, we obtained another dataset from a GWAS meta-analysis in UK Biobank performed by MRC IEU [15] that used binary data and included 5,669 AF/F cases of International Classification of Diseases (ICD)-10 code I48 and 457,341 controls in the European population. This dataset was publicly available from the MRC IEU Open GWAS database [16] and from MR-Base [17], as GWAS-ID of "ukb-b-964." For example, GWAS datasets in the UK Biobank by MRC IEU ("ukb-b-

19953" for body mass index [BMI] and "ukb-b-223" for smoking) were also used in another MR analysis [18]. As the CKDGen GWAS meta-analysis did not include UK Biobank participants [19], there was no apparent sample overlap between the exposure and outcome datasets.

As all data used in the present study were derived from publicly available summary-level GWAS datasets and no individual-level data were used, additional ethical approval and patient consent were not necessary.

## Selection of instrumental variables

In the MR analysis, SNPs from the exposure dataset were used as IVs. IVs must satisfy the following three assumptions: the IVs are associated with the exposure (IV assumption 1); the IVs affect the outcome only via exposure (IV assumption 2); the IVs are not associated with measured or unmeasured confounders (IV assumption 3) [20].

For the causal effect of the exposure on the outcome, the SNPs were selected from the exposure GWAS summary data as IVs by clumping together all SNPs that were associated with the exposure trait at a genome-wide significance threshold ($P < 5.0 \times 10^{-8}$) and were not in linkage disequilibrium (LD) ($r^2 < 0.001$, and distance > 10,000 kb) with the other SNPs. Moreover, the bidirectional MR analysis depends on an assumption that the SNPs used as IVs do not overlap or are not in LD between the exposure and the outcome [9, 21]. When SNPs overlapped or were in LD, the SNPs (if any existed) were excluded from the MR analysis [22]. The summary statistics of each SNP were extracted from both the exposure and outcome datasets and then harmonized. When an exposure SNP was not available in the outcome dataset, we used a proxy SNP (if any existed) with high LD ($r^2 > 0.8$) in combination with the exposure SNP. Palindromic SNPs exhibiting an intermediate minor allele frequency > 0.42 were excluded from the analyses [20].

To evaluate the strength of the exposure IVs, we calculated the F-statistic of each SNP using the following formula: F-statistic = $R^2 \times (N-2)/(1-R^2)$, where $R^2$ is the variance of the phenotype explained by each genetic variant in exposure, and N is the sample size. $R^2$ was calculated using the following formula: $R^2 = 2 \times (Beta)^2 \times EAF \times (1-EAF)/[2 \times (Beta)^2 \times EAF \times (1-EAF) + 2 \times (SE)^2 \times N \times EAF \times (1-EAF)]$, where Beta is the per-allele effect size of the association between each SNP and phenotype, EAF is the effect allele frequency, and SE is the standard error of Beta [23]. IVs with an F-statistic below 10 were considered as weak instruments [24]. Moreover, we used the MR Steiger filtering method that was implemented in the TwoSampleMR package to infer the causal direction of each SNP on the hypothesized exposure and outcome [25]. If a SNP has a causal effect of the exposure on the outcome, the SNP used as an IV should be more predictive of the exposure than the outcome [26]. When a SNP was more predictive of the outcome than the exposure, the SNP (if any existed) was excluded from the MR study as a sensitivity analysis [26, 27].

## Two-sample Mendelian randomization

Wald ratio estimates the causal effect for each IV, and this value was calculated as the ratio of Beta for the corresponding SNP in the outcome dataset divided by Beta for the same SNP in the exposure dataset [20]. Our main approach was to conduct a meta-analysis of each Wald ratio according to the inverse variance weighted (IVW) method to estimate the overall causal effect of the exposure on the outcome. For the IVW method, we used a multiplicative random-effects model when Cochran's Q statistic (as described below) was significant ($P < 0.05$) [28]. Otherwise, a fixed-effects model was used. Additionally, we conducted sensitivity analyses using the weighted median method, the MR-Egger regression method, the weighted mode method, the MR-PRESSO global and outlier tests, and leave-one-out sensitivity analysis. The

weighted median method provides a valid causal estimate when more than half of the instrumental SNPs satisfy the IV assumptions [29]. The MR-Egger regression method is used to assess horizontal pleiotropy of IVs. When IV assumption 2 is violated, horizontal pleiotropy occurs, and the MR-Egger regression intercept is non-zero with statistical significance [29]. The weighted mode method forms clusters of individual SNPs and estimates the causal effect from the largest cluster [29]. The MR-PRESSO global and outlier tests investigate if there are outlier SNPs that possess variant-specific causal estimates that differ substantially from those of other SNPs [30]. Leave-one-out sensitivity analysis was conducted to assess the reliability of the IVW method by removing each SNP from the analysis and re-estimating the causal effect [30]. Heterogeneity was also measured among the causal estimates across all SNPs in the IVW method by calculating Cochran's Q statistic and the corresponding $P$-value. Low heterogeneity provides more reliability for causal effects [31]. Moreover, among all the SNPs used as IVs for the exposure datasets, we searched for SNPs associated with $P < 5.0 \times 10^{-8}$ with possible pleiotropic effects on other diseases and traits using the web-tool PhenoScanner (version 2) [32, 33].

## Results

### Estimating the causal effect of the risk of AF/F on the change in eGFR

First, we investigated the causal effect of the risk of AF/F on changes in eGFR. The exposure dataset for AF/F included 20 instrumental SNPs with rs651386 excluded by LD clumping. As the 20 SNPs were all identified for the outcome GWAS datasets of eGFR, we used no proxy SNPs. The characteristics of the 20 SNPs are listed in S1 Table in S1 File. The F-statistic of every instrument was > 30, thus indicating that there was no weak instrument bias. In bidirectional two-sample MR study, two sets of instrumental SNPs for both traits should not be in LD with each other [9, 21]. We confirmed that none of the 20 SNPs used as IVs for the AF/F dataset overlapped or were in LD with the 144 SNPs used as IVs for the eGFR dataset (see S2 Table in S1 File). For harmonization, rs7853195 was excluded as it was palindromic ("palindromic" was "TRUE" in S1 Table in S1 File) with intermediate allele frequencies ("ambiguous" was "TRUE" in S1 Table in S1 File). All the remaining 19 SNPs were more predictive of the exposure (the risk of AF/F) than the outcome (the change in eGFR) ("Steiger direction" was "TRUE" in S1 Table in S1 File). The MR results are shown in Table 1, Figs 1 and S1. The IVW method using a multiplicative random-effects model suggested that the risk of AF/F may have

**Table 1. MR results of the effects of AF/F on the change of eGFRcr and the risk of CKD.**

| Exposure traits | Outcome traits | Number of SNPs | IVW method | Weighted median method | MR-Egger regression method | | Weighted mode method | Heterogeneity (IVW) | MR-PRESSO global test | Outlier-corrected IVW |
|---|---|---|---|---|---|---|---|---|---|---|
| | | | Beta | Beta | Beta | Intercept | Beta | Cochran's Q | | Beta |
| | | | (SE) | (SE) | (SE) | (SE) | (SE) | | | (SE) |
| | | | P-value | P-value | P-value | P-value | P-value | P-value | P-value | P-value |
| AF/F | eGFRcr | 19 | -0.0721 | -0.121 | -0.137 | 0.000185 | -0.134 | 43.6 | | -0.0955 |
| | | | (0.0585) | (0.0570) | (0.130) | (0.000328) | (0.0584) | | | (0.038) |
| | | | 0.23 | 0.035 | 0.31 | 0.58 | 0.034 | < 0.001 | 0.002 | 0.012 |
| AF/F | CKD | 19 | 3.23 | 3.86 | 4.62 | -0.00397 | 3.84 | 23.0 | | Not Available |
| | | | (1.01) | (1.45) | (2.51) | (0.00638) | (1.63) | | | |
| | | | 0.001 | 0.008 | 0.084 | 0.54 | 0.03 | 0.19 | 0.22 | |

Abbreviations: AF/F, atrial fibrillation/flutter; CKD, chronic kidney disease; eGFR, serum creatinine-based estimated glomerular filtration rate; IVW, inverse variance weighted; MR, Mendelian randomization; SE, standard error; SNPs, single nucleotide polymorphisms.

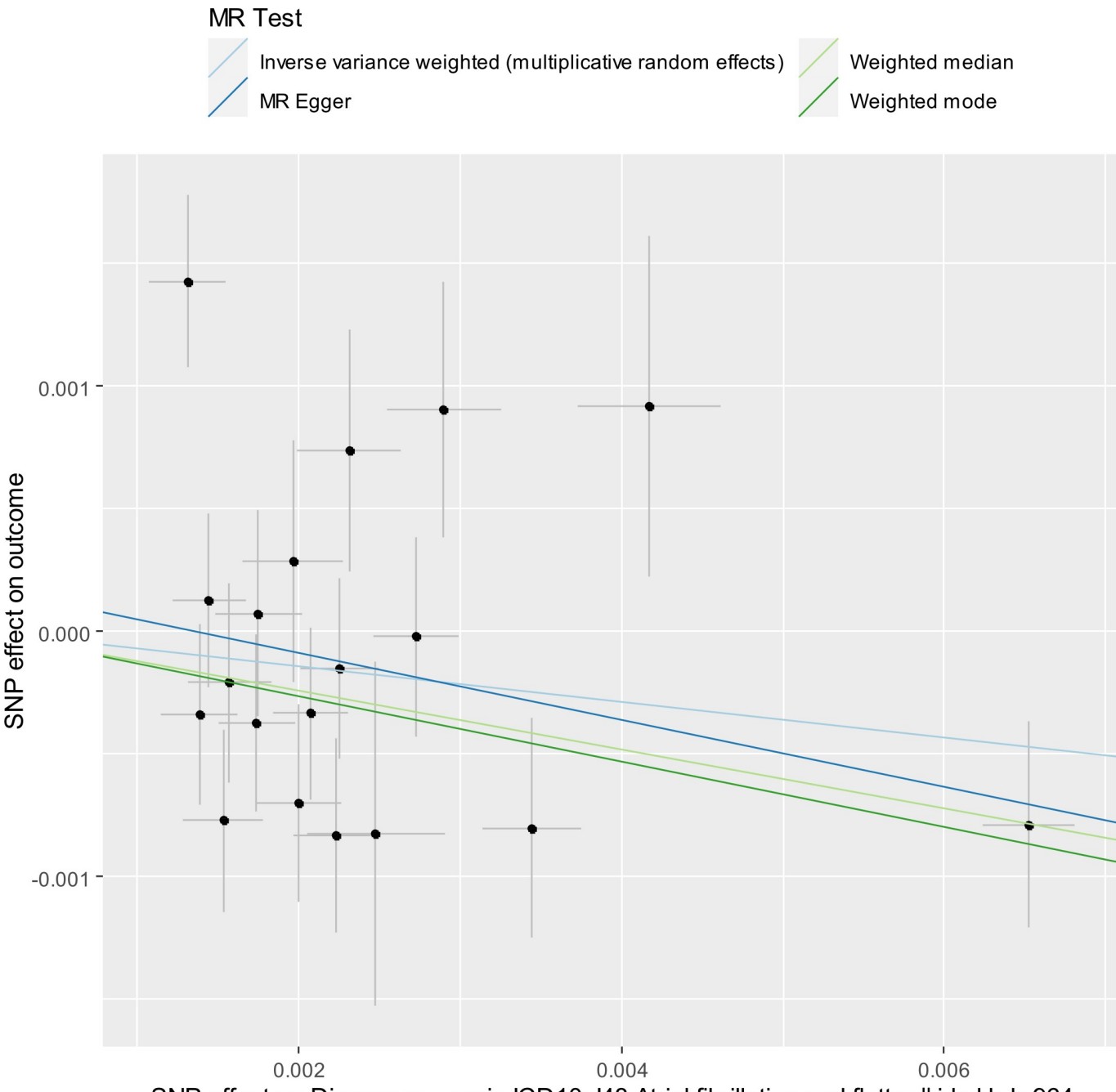

**Fig 1. Scatter plot for estimating the risk of AF/F on the change in eGFR.** Each black point representing a SNP is plotted in relation to the effect size of the SNP on the exposure (x-axis) and on the outcome (y-axis) with corresponding standard error bars. The slope of each line corresponds to the causal estimate using IVW (light blue), weighted median (light green), MR-Egger regression (blue), and weighted mode (green) method.

decreased eGFR (beta for log [eGFR] per log OR of AF/F, -0.0721; SE, 0.0585) [19]; however, the effect was not significant ($P = 0.23$). The results of both the weighted median and weighted mode methods revealed consistent results with suggestive significance. The MR-Egger intercept indicated little evidence of horizontal pleiotropy. MR-PRESSO global and outlier tests indicated that rs796427 was an outlier SNP ($P < 0.0038$), as suggested by the funnel plot

(S2 Fig). When we excluded rs796427 from the IVW method using a fixed-effects model, the risk of AF/F decreased eGFR with statistical significance (Table 1). Here, we used a fixed-effects model for the IVW method, as Cochran's Q statistic for the IVW method indicated low heterogeneity after we excluded rs796427 (Cochran's Q statistic, 24.3; $P$ = 0.11).

## Estimating the causal effect of the risk of AF/F on CKD risk

Next, we investigated the causal effect of the risk of AF/F on the risk of CKD, as the MR analyses indicated a causal effect of the risk of AF/F on the decrease in eGFR when an outlier SNP was excluded. The exposure dataset for AF/F was the same as described above (S1 Table in S1 File). All 19 SNPs were also identified in the outcome GWAS datasets of CKD, and were more predictive of the exposure (the risk of AF/F) than the outcome (the risk of CKD) ("Steiger direction" was "TRUE" in S1 Table in S1 File). The MR results are shown in Table 1, Figs 2 and S1. The IVW method using a fixed-effects model revealed that the risk of AF/F was significantly associated with a higher risk of CKD (OR of CKD per log OR of AF/F, 25.3; 95% coefficient interval [CI], 3.51–183.0; $P$ = 0.001) [19], thus suggesting that the OR of CKD was 9.39 per doubling OR of AF/F (95% CI, 2.39–37.0). For interpretation purposes, we multiplied beta by log (2) (= 0.693) and then exponentiated this value [34]. Other MR methods also provided consistent results although the weighted mode method indicated only suggestive significance (Table 1). Leave-one-out sensitivity analysis demonstrated that the significance disappeared when rs6843082 was excluded from the IVW method (S3 Fig); however, the MR-PRESSO global test revealed that rs6843082 was not an outlier (Table 1). Cochran's Q statistic for the IVW method indicated low heterogeneity and reliability of the causal effect. PhenoScanner identified four SNPs associated with possible pleiotropic effects on other diseases and traits as shown in S1 Table in S1 File (rs35176054 on height, rs6843082 on cardioembolic stroke and ischemic stroke, rs796427 on hand grip strength, arm impedance, and years of educational attainment, and rs879324 on cardioembolic stroke and ischemic stroke). We excluded these four SNPs from the IVW method using a fixed-effects model, and then we obtained a comparable result to that of the original IVW method (OR of CKD per log OR of AF/F, 33.2; 95% CI, 2.07–532.2; $P$ = 0.013). Here, we used a fixed-effects model for the IVW method, as Cochran's Q statistic for the IVW method indicated low heterogeneity after we excluded the four SNPs (Cochran's Q statistic, 17.1; $P$ = 0.25).

## Estimating the reverse causal effect of the change in eGFR on the risk of AF/F

Finally, we investigated the reverse causal effect of the change in eGFR on the risk of AF/F. The exposure dataset of eGFR included 308 instrumental SNPs. A total of 158 SNPs were excluded by LD clumping, and nine SNPs were not identified for the outcome GWAS datasets for AF/F. However, three SNPs were detected as proxy SNPs (rs140124 for rs131263, rs2293579 for rs61897431, and rs147726416 for rs75625374, as shown in S2 Table in S1 File). As a result, 144 SNPs in the exposure dataset of eGFR were used as IVs. The characteristics of the 144 SNPs are listed in S2 Table in S1 File. The F-statistic of every instrument was > 28, thus indicating that there was no weak instrument bias. During harmonization, five SNPs (rs10865189, rs154656, rs55929207, rs8096658, and rs8474) were excluded due to the observation that they were palindromic ("palindromic" was "TRUE" in S2 Table in S1 File) with intermediate allele frequencies ("ambiguous" was "TRUE" in S2 Table in S1 File). All the remaining 139 SNPs were more predictive of the exposure (the change in eGFR) than the outcome (the risk of AF/F) ("Steiger direction" was "TRUE" in S2 Table in S1 File). The MR results are shown in Table 2, Figs 3 and S1. The IVW method using a multiplicative random-effect model

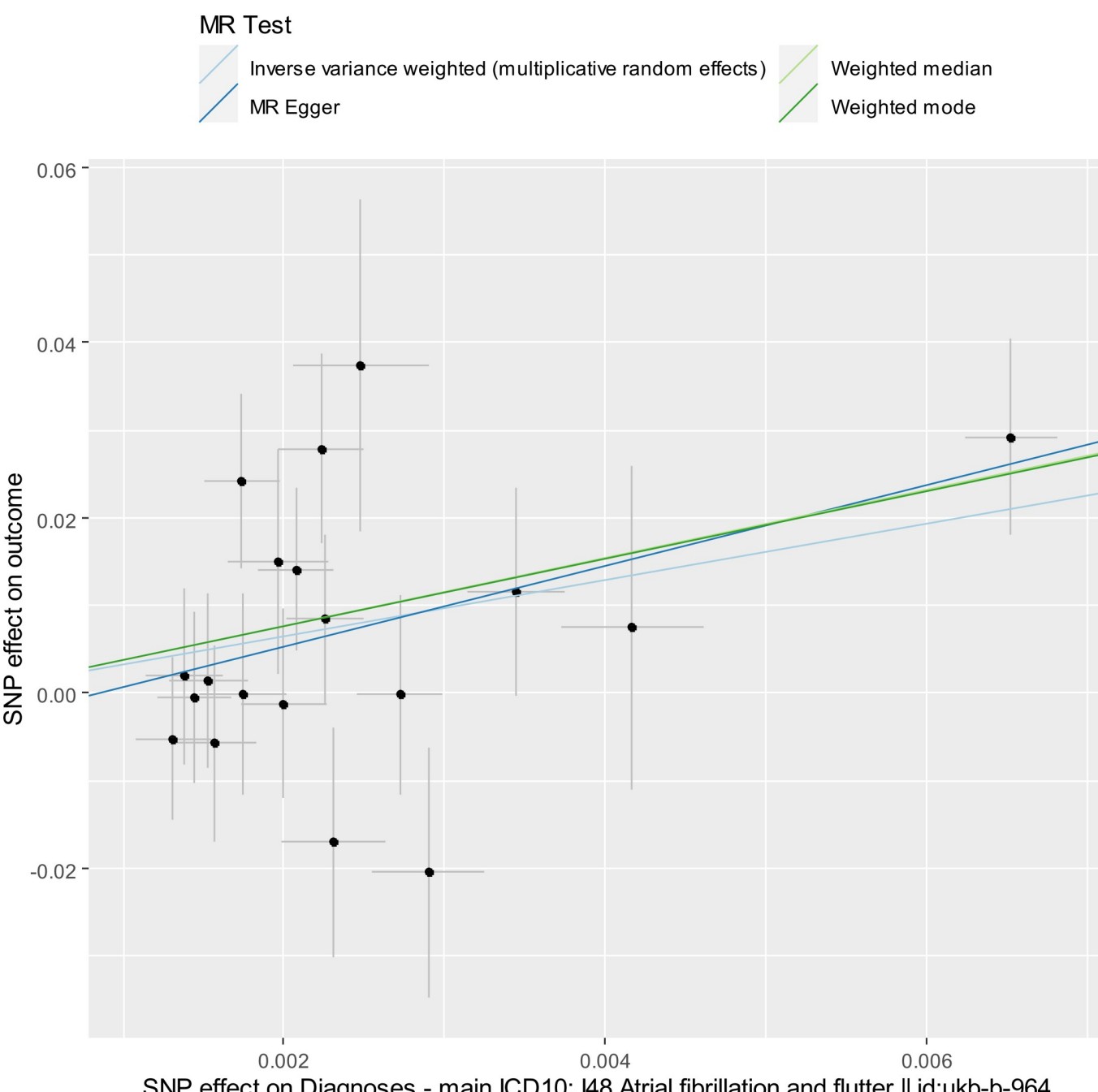

**Fig 2. Scatter plot for estimating the causal effect of the risk of AF/F on the risk of CKD.**

revealed that the change in eGFR was not significantly associated with the risk of AF/F (OR of AF/F per unit change in log[eGFR], 0.996; 95% CI, 0.980–1.013; $P = 0.66$) [11]. None of the other methods showed any causal effects (Table 2). As the MR-PRESSO global and outlier tests indicated that two SNPs were outliers ($P < 0.0278$ for rs3775932, $P < 0.0278$ for rs4656220), as was suggested by the funnel plot (S2 Fig), we excluded them from the IVW method using a multiplicative random-effect model. The result remained insignificant (OR, 0.99967; 95% CI, 0.985–1.015; $P = 0,97$). Moreover, PhenoScanner identified 89 SNPs associated with possible

**Table 2. MR results of the effect of the change of eGFRcr on the risk of AF/F.**

| Exposure traits | Outlier traits | Number of SNPs | IVW method | Weighted median method | MR-Egger regression method | | Weighted mode method | Heterogeneity (IVW) | MR-PRESSO global test | Outlier-corrected IVW |
|---|---|---|---|---|---|---|---|---|---|---|
| | | | Beta | Beta | Beta | Intercept | Beta | Cochran's Q | | Beta |
| | | | (SE) | (SE) | (SE) | (SE) | (SE) | | | (SE) |
| | | | *P*-value | *P*-value | *P*-value | *P*-value | *P*-value | *P*-value | *P*-value | *P*-value |
| eGFRcr | AF/F | 139 | -0.0038 | -0.000766 | 0.0142 | -0.0000699 | -0.00393 | 264.8 | | -0.00033 |
| | | | (0.00852) | (0.00996) | (0.0215) | (0.0000763) | (0.0135) | | | (0.00769) |
| | | | 0.66 | 0.94 | 0.51 | 0.36 | 0.77 | < 0.001 | < 0.001 | 0.97 |

Abbreviations: AF/F, atrial fibrillation/flutter; CKD, chronic kidney disease; eGFR, serum creatinine-based estimated glomerular filtration rate; IVW, inverse variance weighted; MR, Mendelian randomization; SE, standard error; SNPs, single nucleotide polymorphisms.

pleiotropic effects on other diseases and traits among 144 SNPs as shown in S2 Table in S1 File. We excluded all 89 SNPs from the IVW method using a multiplicative random-effect model. The result remained insignificant (OR, 1.005; 95% CI, 0.980–1,031; *P* = 0.69). Here, we used a random-effects model for the IVW method, as Cochran's Q statistic for the IVW method indicated high heterogeneity after we excluded the 89 SNPs (Cochran's Q statistic, 100.4; *P* = 0.015).

## Discussion

To the best of our knowledge, this is the first MR analysis to estimate causal associations between the risk of AF/F and the change in eGFR or the risk of CKD, and we made two novel discoveries in the European population. First, our study suggested a causal effect of the risk of AF/F on the risk of CKD. Second, the causal effect of the change in eGFR on the risk of AF/F was unlikely. However, careful attention must be paid to interpreting these results as our sample size of the AF/F dataset was relatively small.

Several observational studies have reported a higher prevalence of AF in a dose-dependent manner as kidney function decreased [3–8]. For example, the cross-sectional CRIC study in the US reported that the OR of prevalent AF was 1.35 (95% CI, 1.13–1.62) in subjects with eGFR $<$ 45 ml/min/1.73m$^2$ compared to those with eGFR $>$ 45 ml/min/1.73m$^2$ [3]. The cross-sectional REGARDS study in the US reported an OR of AF that was 2.86 (95% CI, 1.38–5.92) in CKD Stage 4–5 compared to no CKD [4]. However, few observational studies have reported a causal effect of the prevalence of AF on the risk of CKD. The prospective Niigata preventive medicine study in Japan reported that among subjects without hypertension or diabetes during a mean follow-up of 5.9 years, the development of kidney dysfunction was 16.6 incidences per 1000 person-years (95% CI, 13.0–20.2) in subjects with baseline AF, and only 5.2 (95% CI, 5.0–5.3) in those without [5]. Our MR results were consistent with those of the Niigata preventive medicine study regarding the causal association of AF prevalence with kidney dysfunction. However, our MR study did not support the causal effect of kidney dysfunction on AF prevalence that was previously reported by several observational studies. Although our sample size of the AF/F dataset was relatively small, this discrepancy may be partly due to the knowledge that observational studies lacking randomization designs are generally prone to bias resulting from various factors including confounders and reverse causations [9]. The possibility of confounding the association between AF and CKD is always present in observational studies [35]. Consistent with our MR results suggesting a lack of causal effect of the eGFR decrease on the risk of AF/F, an MR study revealed that the urine albumin adjusted for creatinine as a proxy

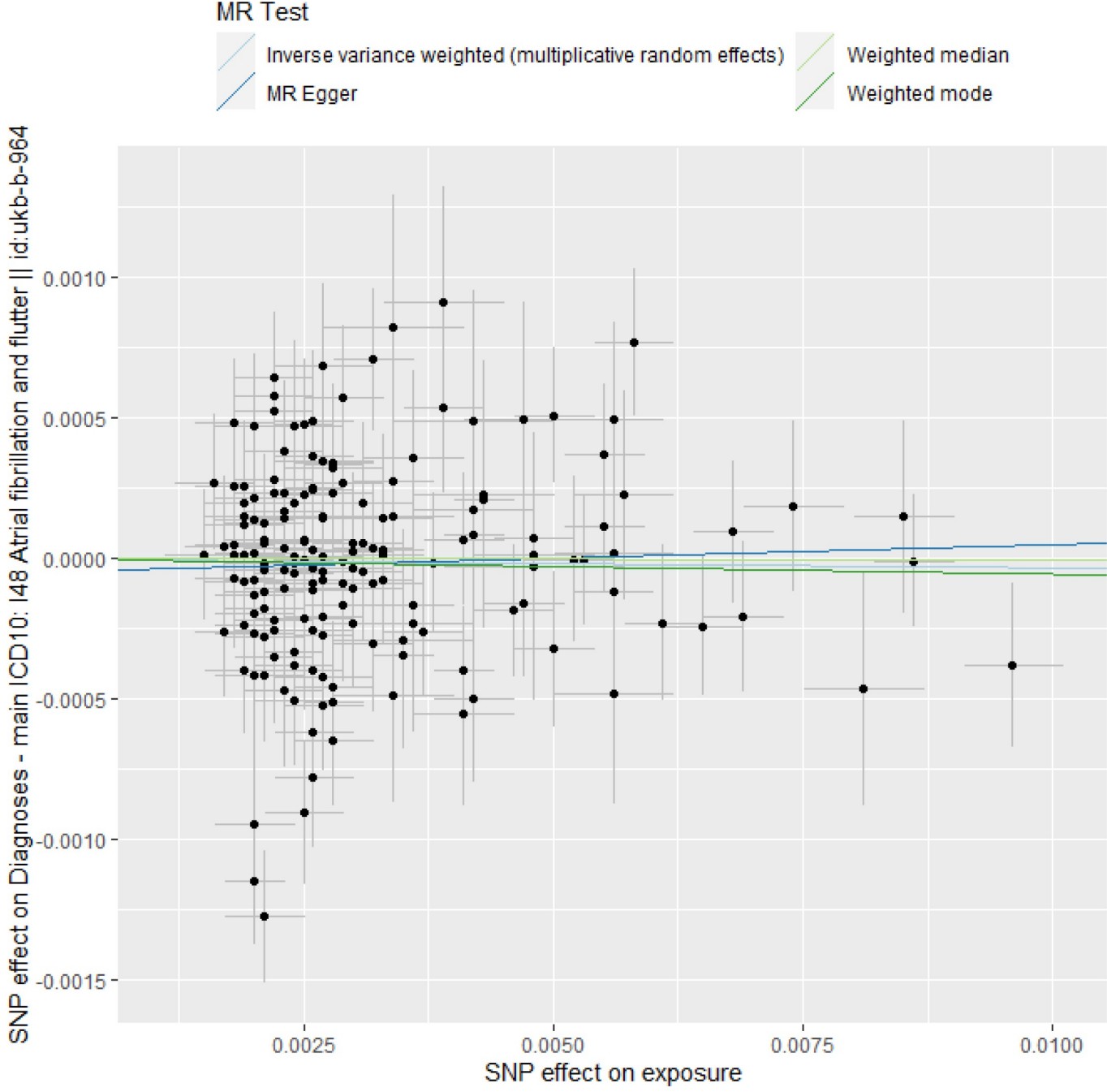

**Fig 3. Scatter plot for estimating the causal effect of the change in eGFR on the risk of AF/F.**

for kidney function did not exert a significant causal effect on the outcome of AF (beta, 0.105; 95% CI, -0.064–0.274; *P* = 0.23) [36].

The mechanisms by which AF/F may cause kidney dysfunction remain unknown. Several risk factors are shared between AF and CKD, including elevated inflammation and an activated renin-angiotensin-aldosterone system (RAAS) [35]. Kidney dysfunction can occur when AF/F causes systemic inflammation and RAAS activation, but the reverse may also be true. As a possible mechanism, a decrease in cardiac output due to AF/F may cause pre-renal failure

leading to chronic kidney dysfunction. Additionally, thromboembolism due to AF/F may cause renal artery occlusion [5]. Our search using PhenoScanner determined that two of our exposure SNPs of AF/F (rs6843082 and rs879324) also effected cardioembolic stroke traits ($P < 5.0 \times 10^{-8}$), which is one of the AF/F comorbidities [37]. This supports the idea that AF/F could cause kidney dysfunction via thromboembolism. However, further studies are required to elucidate the precise mechanisms involved.

Our MR estimate scale of OR of CKD per doubling OR of AF/F may be too large and the corresponding 95% CI width was very wide (OR, 9.39; 95%CI, 1.99–44.2). One possible reason for this observation was the relatively small sample size of the AF/F dataset that was one of the major limitations of this study, as we could not use the largest GWAS meta-analysis due to substantial sample overlap. In general, the genetic instruments used in MR studies can estimate the lifetime effect, and this may explain the larger estimates compared to observational studies [38]. We believe that given that the primary aim of our MR study was to assess if the exposure had a causal effect on the outcome, estimating the size of the causal effect was less important [30]. On the other hand, we could not detect, if any existed, the causal effect of the change in eGFR on the risk of AF/F probably because the scale of the OR was very small. The F-statistic of every SNP > 28 indicated that there was no weak instrument bias (S2 Table in S1 File).

Selection bias was also one of the major limitations of this study. Our MR Steiger filtering method inferred the causal direction of all 19 SNPs used as IVs for the AF/F datasets on the exposure (the risk of AF/F) and outcome (the risk of CKD). However, our instrumental SNPs for the risk of AF/F were selected from the same GAWS dataset as used for subsequent analyses, as is often the case with typical MR studies. We did not use three-sample MR design (three non-overlapping GWASs: selection dataset, exposure dataset, outcome dataset) in the present study [39]. Then, our instrumental SNPs could not be regarded as random samples because they were selected at a genome-wide significant threshold ($P < 5.0 \times 10^{-8}$) [40]. The double use of the same sample for SNP/IV selection and estimation was subject to selection bias and horizontal pleiotropy that could invalid the causality, resulting in inflated type 1 error rates and excessive false positives in the MR Steiger method [21, 40, 41]. Therefore, careful interpretation of the causality is warranted in our MR study.

There are other limitations in our study. The causal effect of the risk of AF/F on the decrease in eGFR was indicated only when an outlier SNP was excluded. The GWAS dataset "ukb-b-964" with ICD-10 code I48 included cases with atrial flutter in addition to AF, although so did the DiscovEHR study that was included in Nielsen's GWAS [13]. Our analysis was based on populations of European ancestry, and the findings are unlikely generalized to other populations. Conversely, the lack of possible sample overlap between the exposure and outcome datasets was a strength of our study that allowed us to avoid substantial bias.

In conclusion, our MR analyses suggested a causal effect of the risk of AF/F on the decrease in eGFR and revealed a causal effect of the risk of AF/F on the risk of CKD. Conversely, the reverse causal effect of the decrease of eGFR on the risk of AF/F was unlikely. However, careful interpretation and further studies are warranted, as the sample size was relatively small and selection bias was possible.

## Supporting information

**S1 Checklist. STREGA reporting recommendations, extended from STROBE statement.** (DOC)

**S1 Fig. Forrest plots.** (a) Forrest plot for estimating the risk of AF/F on the change in eGFR. (b) Forrest plot for estimating the causal effect of the risk of AF/F on the risk of CKD. (c)

Forrest plot for estimating the causal effect of the change in eGFR on the risk of AF/F. Each black point represents the causal estimate of each SNP on the outcome per increase in the exposure, and red points show the combined causal estimates using IVW and MR-Egger regression methods with horizontal lines denoting 95% confidence intervals.
(PPTX)

**S2 Fig. Funnel plots.** (a) Funnel plot for estimating the risk of AF/F on the change in eGFR. (b) Funnel plot for estimating the causal effect of the risk of AF/F on the risk of CKD. (c) Funnel plot for estimating the causal effect of the change in eGFR on the risk of AF/F. Each black point representing an SNP is plotted in relation to the estimate of the exposure on the outcome (x-axis) and the inverse of the standard error (y- axis). Vertical lines show the combined causal estimates using IVW (light blue) and MR- Egger regression (blue) methods.
(PPTX)

**S3 Fig. Leave-one-out sensitivity analyses.** (a) Leave-one-out sensitivity analysis for estimating the risk of AF/F on the change in eGFR. (b) Leave-one- out sensitivity analysis for estimating the causal effect of the risk of AF/F on the risk of CKD. (c) Leave-one-out sensitivity analysis for estimating the causal effect of the change in eGFR on the risk of AF/F. Each black point represents the combined causal estimates on the outcome per increase in the exposure using IVW methods with horizontal lines denoting 95% confidence intervals after removing the corresponding SNP from the analysis.
(PPTX)

**S1 File. Contains all the supporting tables.**
(XLSX)

## Acknowledgments

We would like to thank the CKDGen consortium and MRC IEU UK Biobank GWAS pipeline for making the GWAS datasets publicly available.

## Author Contributions

**Conceptualization:** Masahiro Yoshikawa.

**Data curation:** Masahiro Yoshikawa.

**Formal analysis:** Masahiro Yoshikawa.

**Investigation:** Masahiro Yoshikawa.

**Methodology:** Masahiro Yoshikawa.

**Project administration:** Masahiro Yoshikawa.

**Resources:** Masahiro Yoshikawa, Kensuke Asaba.

**Software:** Masahiro Yoshikawa.

**Validation:** Masahiro Yoshikawa, Kensuke Asaba, Tomohiro Nakayama.

**Visualization:** Masahiro Yoshikawa.

**Writing – original draft:** Masahiro Yoshikawa.

**Writing – review & editing:** Masahiro Yoshikawa, Kensuke Asaba, Tomohiro Nakayama.

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
