## [Decision Letter · Decision Letter 0]

23 Jun 2021

PONE-D-21-11069

Causal effect of atrial fibrillation/flutter on chronic kidney disease: A two-sample Mendelian randomization study

PLOS ONE

Dear Dr. Yoshikawa,

Thank you for submitting your manuscript to PLOS ONE. After careful consideration, we feel that it has merit but does not fully meet PLOS ONE’s publication criteria as it currently stands. Therefore, we invite you to submit a revised version of the manuscript that addresses the points raised during the review process.

After external review, both reviewers raised some concerns about the methodology. Please respond to each of the comments and revise accordingly.

We look forward to receiving your revised manuscript.

Kind regards,

Jie V Zhao

Academic Editor

PLOS ONE

Journal Requirements:

3. Please include captions for *all* your Supporting Information files at the end of your manuscript, and update any in-text citations to match accordingly. Please see our Supporting Information guidelines for more information: http://journals.plos.org/plosone/s/supporting-information.

Reviewers' comments:

Reviewer's Responses to Questions

**Comments to the Author**

1. Is the manuscript technically sound, and do the data support the conclusions?

Reviewer #1: No

Reviewer #2: Partly

2. Has the statistical analysis been performed appropriately and rigorously? 

Reviewer #1: No

Reviewer #2: No

3. Have the authors made all data underlying the findings in their manuscript fully available?

Reviewer #1: Yes

Reviewer #2: Yes

4. Is the manuscript presented in an intelligible fashion and written in standard English?

Reviewer #1: Yes

Reviewer #2: Yes

5. Review Comments to the Author

Reviewer #1: This research studies the causal effect of atrial fibrillation/flutter on chronic kidney

disease. It is a standard application of two-sample Mendelian randomization method implemented in the R package TwoSampleMR.

The manuscript is well written. The description is clear and well laid out.

My main concern is on the way the IV SNPs are selected. A recent study has established that using the same data that are used to select the IVs in the subsequent analysis can lead to biased results (Wang and Han, 2021). Any comments on how your results are affected by SNPs selection?

Furthermore, in the original paper by Bowden et al (2015) and an extension by Sue and Pan (2020), the selection of IV SNPs are such that they are significant in both the exposure GWAS AND the trait GWAS. However, your IV SNPs seem to be selected based only on the exposure GWAS, not on the trait GWAS. Am I correct? If I am, do you have any comments?

Some minor comments:

1. In lines 118-119, the R^2 is expressed as a fraction. The term "EAFx(1-EAF)" appears in both the numerator and the denominator. Shouldn't term be cancelled out?

2. The figures in the text and the appendices are difficult to read.

References:

Wang, K. and Han, S., 2021. Effect of selection bias on two sample summary data based Mendelian randomization. Scientific reports, 11(1), pp.1-8.

Bowden, J., Davey Smith, G. and Burgess, S., 2015. Mendelian randomization with invalid instruments: effect estimation and bias detection through Egger regression. International journal of epidemiology, 44(2), pp.512-525.

Xue, H. and Pan, W., 2020. Inferring causal direction between two traits in the presence of horizontal pleiotropy with GWAS summary data. PLoS genetics, 16(11), p.e1009105.

Reviewer #2: Causal effect of atrial fibrillation/flutter on chronic kidney disease: A two-sample Mendelian randomization study

By Masahiro Yoshikawa and Kensuke Asaba

Reviewer Comments

The Authors conducted a Mendelian Randomization (MR) study to assess causality of atrial fibrillation/flutter (AF/F) and estimated glomerular filtration rate (eGFR)/chronic kidney disease (CKD), both directions. While state-of-the-art methods are essentially applied I have serious issues about methods and their presentations. Without their resolution this presentation could be misunderstood and incorrect procedure promoted. Still, the results might be valid but to ensure this up to a certain point proper methodology needs to be applied/reported. In the following I distinguish major and minor points:

Major points:

1 Data source: CKDGen GWAS on eGFR

In this large meta-analysis, contributing studies derived residuals for log(eGFR) which then were used as outcome in the GWAS. It is incorrect to assume that this trait has SD of 1. This assumption was already incorrectly made by another study on which the Authors rely. Please revise.

2 Palindromic SNPs

Authors define palindromic SNPs with minor allele frequency (MAF) >0.42 and excluded thus some of the otherwise eligible SNPs. However, the Authors overlooked that a palindromic SNPs is also one with an effect allele frequency (EAF) – as presented in Supplementary Tables 1 and 2 - of, for example, 0.55. Since there are several SNPs with an EAF in the range of 0.5 to 0.58, exclusion of SNPs is incomplete. The analysis should be thus revised.

3 Estimation of R² and F-statistics

Especially regarding R², I am not familiar with this presentation of formula. The reference provided by the Authors is also only an application that in turn point to a reference of Palmer et al 2012. Tracing that paper, however, does not provide the explanation on the formula. While the formula might be correct (I know presentations that look similar to a certain degree), it would be good to see who this formula proposed and how it was derived.

The main reason for my search was that a proper R² incorporates the variance of the outcome. In case such as in a 2-sample MR study the variance is often not available why usually some simplifying assumption such as SD=1 is applied. Since this is mostly likely not true for eGFR (see my first comment) and since I cannot deduce any other approximation of the variance of the outcome in the presented formula, I am wondering how reliable the estimates of R² and thus of the F-statistics are.

As – in the worst case – the strengths of instruments could be overestimated, this aspect needs clarification and discussion.

4 Post-hoc power calculations

As discussed by many authors/statisticians (e.g. Dziak et al 2020, PMID: 32523323), post-hoc power calculations as presented in this paper are not valid. This part needs to be deleted and the topic differently approached.

Minor points:

1 Since both directions are assessed and the standard phrasing is that of a “bidirectional” MR study it should be stated in the title as well as in the introduction.

2 Please provide phenotype definitions as used in the underlying studies.

3 Please add further information on SNPs in Supplementary Tables 1 and 2, e.g., marking palindromic SNPs, outliers and otherwise noticeable SNPs as well as adding Phenoscanner information.

4 Please do Phenoscanner look up for all SNPs. Please be cautious in your wording of “unrelated traits”. Sometimes relation may not be so obvious.

5 In presentation of results on the evaluation of the causal relationship of AF/F on eGFR, the Authors state that they used a multiplicative random effects model because of observed heterogeneity. Since the results on IVW estimate is presented above, this statement needs clarification if the presented result is already from multiplicative random effects model, also in contrast to all other presented results from IVW analysis. Was this the only instance?

6 The Authors state in the Discussion that estimates may be too large and CIs were too wide and these present a major limitation of the study. However, the reported results present maybe a consequence of some limitations but do not present a limitation themselves. Please revise.

6. PLOS authors have the option to publish the peer review history of their article (what does this mean?). If published, this will include your full peer review and any attached files.

Reviewer #1: No

Reviewer #2: No

---

## [Author Response · Author response to Decision Letter 0]

20 Aug 2021

Dear Dr. Jie V Zhao, 

We would like to sincerely thank you very much for the time and effort you have dedicated to providing insightful feedback on ways to strengthen our manuscript entitled “Causal effect of atrial fibrillation/flutter on chronic kidney disease: A two-sample (bidirectional) Mendelian randomization study” under consideration for publication in PLOS ONE.

First, we would like to sincerely apologize for our major mistake and to correct it. 

Among 178 SNPs used as IVs for the change of eGFR in the original version of Supplementary Table 2, a total of 34 SNPs were not associated with eGFR at a genome-wide significance threshold (P = 5E-08) and they should have been excluded from the subsequent analyses (for example, rs10197255, P = 4.7E-07 in the original version of Supplementary Table 2).

To satisfy IV assumption 1 (the IVs are associated with the exposure), we should have selected SNPs as IVs that were associated with the exposure trait with P < 5.0×10-8 from the exposure GWAS summary data.

This error was due to the default setting of the clump_data function in the TwoSampleMR package (https://mrcieu.github.io/TwoSampleMR/reference/clump_data.html).

We originally selected the SNPs used as IVs for the change of eGFR from Supplementary Table 4 (ST4) in the GWAS study by CKDgen (Please refer to Nat Genet. 2019;51: 957-972, https://www.ncbi.nlm.nih.gov/pmc/articles/PMC6698888/).

Some of the 308 SNPs listed in ST4 were not associated with eGFR at a genome-wide significance threshold (that is, P > 5.0×10-8) based on populations of European ancestry.

And the default setting of the clump_data() function was as follows: 

clump_data(dat, 

 clump_kb = 10000,

 clump_r2 = 0.001,

 clump_p1 = 1,

 clump_p2 = 1,

 pop = "EUR"

)

Therefore, the 34 wrong SNPs with P > 5.0×10-8 were not excluded from our study by LD clumping. 

(Please refer to the paragraph of "Estimating the reverse causal effect of the change in eGFR on the risk of AF/F" in the original version of the Result session.)

Now, we have changed the setting of the clump_data() function as follows:

clump_data(dat,

 clump_kb = 10000,

 clump_r2 = 0.001,

 clump_p1 = 5E-08,

 clump_p2 = 1,

 pop = "EUR"

) 

And we have excluded a total of 158 SNPs by LD clumping including the 34 wrong SNPs.

(Please refer to the paragraph of "Estimating the reverse causal effect of the change in eGFR on the risk of AF/F" in the revised version of the Result session.)

Although nine SNPs have not been identified for the outcome GWAS datasets for AF/F, three SNPs were detected as proxy SNPs.

Then, we have used a total of 144 SNPs for estimating the causal effect of the change in eGFR on the risk of AF/F. (144 SNPs = 178 SNPs - 34 wrong SNPs.)

The 144 SNPs have been listed correctly in the revised version of Supplementary Table 2. 

Fortunately, the result of causality has been unchanged (the change in eGFR was not significantly associated with the risk of AF/F) though odds ratio, 95% CI, and P-value have been slightly changed.

Incorrect: OR, 0.997; 95% CI, 0.981-1.012; P = 0.67.

Correct: OR, 0.996; 95% CI, 0.980-1.013; P = 0.66. 

(Please see lines 246-247 in the revised manuscript.)

Incorrect: OR, 1.00081; 95% CI, 0.987-1.015; P = 0.91.

Correct: OR, 0.99967; 95% CI, 0.985-1.015; P = 0.97.

(Please see lines 251-252 in the revised manuscript.)

Please also refer to Table 2 in the revised manuscript.

Second, we would like to correct one more mistake.

IV assumption 1 was not "the IVs are associated with the outcome" but "the IVs are associated with the exposure". 

(Please see line 105 in the revised manuscript.)

The errata above are also described in the response to each Reviewer. 

Third, we would like to invite Dr. Tomohiro Nakayama to be a co-author of our paper. 

He contributed substantially to our work by interpreting data, revising the manuscript, and reviewing it critically during this revision process. 

Finally, our responses to each Reviewer’s comment are described below in a point-to-point manner. 

In addition, we have submitted the revised manuscript as well as the document file with tracked changes to highlight the revisions in red color.

We hope that our manuscript will be acceptable for publication in PLOS ONE.

Sincerely, 

Masahiro Yoshikawa, M.D., Ph.D. 

Corresponding author.

Address; Division of Laboratory Medicine, Department of Pathology and Microbiology, Nihon University School of Medicine, Oyaguchi-kamicho 30-1, Itabashi, Tokyo, Japan. 

Phone; 81-3-3972-8111

e-mail: myosh-tky@umin.ac.jp

 

To Reviewer #1

We would like to sincerely thank you very much for the time and effort you have dedicated to providing insightful feedback on ways to strengthen our manuscript entitled “Causal effect of atrial fibrillation/flutter on chronic kidney disease: A two-sample (bidirectional) Mendelian randomization study” under consideration for publication in PLOS ONE.

First, we would like to sincerely apologize for our major mistake and to correct it. 

Among 178 SNPs used as IVs for the change of eGFR in the original version of Supplementary Table 2, a total of 34 SNPs were not associated with eGFR at a genome-wide significance threshold (P = 5E-08) and they should have been excluded from the subsequent analyses (for example, rs10197255, P = 4.7E-07 in the original version of Supplementary Table 2).

To satisfy IV assumption 1 (the IVs are associated with the exposure), we should have selected SNPs as IVs that were associated with the exposure trait with P < 5.0×10-8 from the exposure GWAS summary data.

This error was due to the default setting of the clump_data function in the TwoSampleMR package (https://mrcieu.github.io/TwoSampleMR/reference/clump_data.html).

We originally selected the SNPs used as IVs for the change of eGFR from Supplementary Table 4 (ST4) in the GWAS study by CKDgen (Please refer to Nat Genet. 2019;51: 957-972, https://www.ncbi.nlm.nih.gov/pmc/articles/PMC6698888/).

Some of the 308 SNPs listed in ST4 were not associated with eGFR at a genome-wide significance threshold (that is, P > 5.0×10-8) based on populations of European ancestry.

And the default setting of the clump_data() function was as follows: 

clump_data(dat, 

 clump_kb = 10000,

 clump_r2 = 0.001,

 clump_p1 = 1,

 clump_p2 = 1,

 pop = "EUR"

)

Therefore, the 34 wrong SNPs with P > 5.0×10-8 were not excluded from our study by LD clumping. 

(Please refer to the paragraph of "Estimating the reverse causal effect of the change in eGFR on the risk of AF/F" in the original version of the Result session.)

Now, we have changed the setting of the clump_data() function as follows:

clump_data(dat,

 clump_kb = 10000,

 clump_r2 = 0.001,

 clump_p1 = 5E-08,

 clump_p2 = 1,

 pop = "EUR"

) 

And we have excluded a total of 158 SNPs by LD clumping including the 34 wrong SNPs.

(Please refer to the paragraph of "Estimating the reverse causal effect of the change in eGFR on the risk of AF/F" in the revised version of the Result session.)

Although nine SNPs have not been identified for the outcome GWAS datasets for AF/F, three SNPs were detected as proxy SNPs.

Then, we have used a total of 144 SNPs for estimating the causal effect of the change in eGFR on the risk of AF/F. (144 SNPs = 178 SNPs - 34 wrong SNPs.)

The 144 SNPs have been listed correctly in the revised version of Supplementary Table 2. 

Fortunately, the result of causality has been unchanged (the change in eGFR was not significantly associated with the risk of AF/F) though odds ratio, 95% CI, and P-value have been slightly changed.

Incorrect: OR, 0.997; 95% CI, 0.981-1.012; P = 0.67.

Correct: OR, 0.996; 95% CI, 0.980-1.013; P = 0.66. 

(Please see lines 246-247 in the revised manuscript.)

Incorrect: OR, 1.00081; 95% CI, 0.987-1.015; P = 0.91.

Correct: OR, 0.99967; 95% CI, 0.985-1.015; P = 0.97.

(Please see lines 251-252 in the revised manuscript.)

Please also refer to Table 2 in the revised manuscript.

Second, we would like to correct one more mistake.

IV assumption 1 was not "the IVs are associated with the outcome" but "the IVs are associated with the exposure". 

(Please see line 105 in the revised manuscript.)

Our responses to the Reviewer’s comments

#1. My main concern is on the way the IV SNPs are selected. A recent study has established that using the same data that are used to select the IVs in the subsequent analysis can lead to biased results (Wang and Han, 2021). Any comments on how your results are affected by SNPs selection?

Thank you very much for your suggestion.

We have carefully read the study by Wang and Han. They considered the two-sample MR Steiger method in their study and referred to the study by Hemani et al [26]. 

Hemani et al. introduced the method using the R programming language and implemented it in the MR-Base [26].

Some studies [27, 28] performed the MR Steiger filtering method with the reference to the study by Hemani et al [26]. 

(https://mrcieu.github.io/TwoSampleMR/reference/steiger_filtering.html).

Therefore, we have performed the MR Steiger filtering method using steiger_filtering function in the TwoSampleMR package to infer the causal direction of each SNP on the hypothesized exposure and outcome. 

As a result, all 19 SNPs used as IVs for the risk of AF/F were more predictive of the exposure (the risk of AF/F) than the outcome (the change in eGFR) (“Steiger direction” was “TRUE” in the revised version of Supplementary Table 1).

(Please also see lines 127-133, 174-176, 207-208, and 242-244 in the revised manuscript.)

However, as Reviewer pointed out, using the same data as used to select the IVs in the subsequent analysis can lead to biased results.

Our instrumental SNPs for the risk of AF/F were selected from the same GAWS dataset as used for subsequent analyses. Then, our instrumental SNPs could not be regarded as random samples because they were selected at a genome-wide significant threshold (P < 5.0×10-8) [41].

This could cause selection bias and invalid the causality, resulting in inflated type 1 error rates and excessive false positives in the MR Steiger method.

Therefore, selection bias is one of major limitations, and careful interpretation of the causality is warranted in our study.

We have reflected these comments, in addition to the next comments, to the revised manuscript. Please refer to the next point #2.

(References)

26. Hemani G, Tilling K, Davey Smith G. Orienting the causal relationship between imprecisely measured traits using GWAS summary data. PLoS Genet. 2017;13: e1007081.

27. Treur JL, Demontis D, Smith GD, Sallis H, Richardson TG, Wiers RW, et al. Investigating causality between liability to ADHD and substance use, and liability to substance use and ADHD risk, using Mendelian randomization. Addict Biol. 2021;26: e12849. 

28. Zheng J, Brion MJ, Kemp JP, Warrington NM, Borges MC, Hemani G, et al. The Effect of Plasma Lipids and Lipid-Lowering Interventions on Bone Mineral Density: A Mendelian Randomization Study. J Bone Miner Res. 2020;35: 1224-1235.

41. Wang K, Han S. Effect of selection bias on two sample summary data based Mendelian randomization. Sci Rep. 2021;11: 7585.

#2. Furthermore, in the original paper by Bowden et al (2015) and an extension by Sue and Pan (2020), the selection of IV SNPs are such that they are significant in both the exposure GWAS AND the trait GWAS. However, your IV SNPs seem to be selected based only on the exposure GWAS, not on the trait GWAS. Am I correct? If I am, do you have any comments?

Thank you very much for providing these insights, and as Reviewer pointed out, our instrumental SNPs were selected based only on the exposure GWAS.

As Xue and Pan described in their study [22], bi-directional MR works depends on one critical, most often unknown, assumption: two sets of valid SNPs/IVs used in the two directions. That is, if an SNP primarily influences X, but influences Y only through X, then it should be used as an IV to infer X → Y in the first step; at the same time, it cannot be used as an IV to infer Y → X.

Similarly, as Davey Smith G and Hemani G [21] described, 

“In bi-directional MR, if trait A causes trait B, then the instrument, ZA, will be associated with both A and B. 

However, a second instrument specific to trait B, ZB, will be associated with trait B, and not with trait A. 

This method is only valid on the condition that the two instruments are not marginally associated with each other (e.g. there is no LD between instruments for A and B).”

Then, a study [23] excluded SNPs used as IVs for one trait that were in LD with the significant SNPs for the other trait by referring to the study by Davey Smith G and Hemani G [22].

Therefore, we checked whether our instrumental SNPs for the risk of AF/F overlapped or were in LD with those for the change in eGFR, and confirmed that there were no overlap or LD between the two sets of SNPs/IVs. 

(Please see lines 112-115 and 169-172 in the revised manuscript.)

However, as Reviewer pointed out, our instrumental SNPs for the risk of AF/F were selected from the same GAWS dataset as used for subsequent analyses. We did not use three-sample MR design (three non-overlapping GWAS: selection dataset, exposure dataset, outcome dataset) [40].

As Xue and Pan described [22], the double use of the same sample for SNP/IV selection and estimation can cause possible selection bias due to wide-spread horizontal pleiotropy. Similarly, Bowden et al. [42] described that, if genetic variants are chosen due to their association with the exposure in the dataset under analysis, then the association with the exposure is likely to be overestimated, and the association with the outcome could also then be overestimated due to confounding.

Therefore, also in our study, this double use of the same sample for SNP/IV selection and estimation was subject to selection bias and horizontal pleiotropy that could invalid the causality, resulting in inflated type 1 error rates and excessive false positives.

We have reflected these comments, in addition to the previous comments (please refer to the previous point #1), to the revised manuscript as follows. 

Selection bias was also one of the major limitations of this study. Our MR Steiger filtering method inferred the causal direction of all 19 SNP used as IVs for the AF/F datasets on the exposure (the risk of AF/F) and outcome (the risk of CKD). However, our instrumental SNPs for the risk of AF/F were selected from the same GAWS dataset as used for subsequent analyses, as is often the case with typical MR studies. We did not use three-sample MR design (three non-overlapping GWAS: selection dataset, exposure dataset, outcome dataset) in the present study [40]. Then, our instrumental SNPs could not be regarded as random samples because they were selected at a genome-wide significant threshold (P < 5.0×10-8) [41]. The double use of the same sample for SNP/IV selection and estimation was subject to selection bias and horizontal pleiotropy that could invalid the causality, resulting in inflated type 1 error rates and excessive false positives in the MR Steiger method [22, 41, 42]. Therefore, careful interpretation of the causality is warranted in our MR study.

(Please see lines 319-331 in the revised manuscript.)

Please also refer to lines 31 and 344 in the revised manuscript.

(References)

21. Davey Smith G, Hemani G. Mendelian randomization: genetic anchors for causal inference in epidemiological studies. Hum Mol Genet. 2014;23: R89-98. 

22. Xue H, Pan W. Inferring causal direction between two traits in the presence of horizontal pleiotropy with GWAS summary data. PLoS Genet. 2020;16: e1009105. 

23. Wang K, Ding L, Yang C, Hao X, Wang C. Exploring the Relationship Between Psychiatric Traits and the Risk of Mouth Ulcers Using Bi-Directional Mendelian Randomization. Front Genet. 2020;11: 608630.

40. Zhao Q, Chen Y, Wang J, Small DS. Powerful three-sample genome-wide design and robust statistical inference in summary-data Mendelian randomization. Int J Epidemiol. 2019;48: 1478–1492. 

41. Wang K, Han S. Effect of selection bias on two sample summary data based Mendelian randomization. Sci Rep. 2021;11: 7585. 

42. Bowden J, Davey Smith G, Burgess S. Mendelian randomization with invalid instruments: effect estimation and bias detection through Egger regression. Int J Epidemiol. 2015;44: 512-25.

Some minor comments:

#3. In lines 118-119, the R^2 is expressed as a fraction. The term "EAFx(1-EAF)" appears in both the numerator and the denominator. Shouldn't term be cancelled out?

We agree with Reviewer on this point. 

In fact, we cancelled out 2×EAF×(1-EAF) when we calculated R2. 

As Reviewer #2 pointed out, our reference was inappropriate. This formula was originally derived from the study by Sim H et al. (Please refer to S1 Text in PLoS One. 2015;10: e0120758. https://www.ncbi.nlm.nih.gov/pmc/articles/PMC4405269/) 

#4. The figures in the text and the appendices are difficult to read.

We would like to apologize for any inconvenience we have caused. We have made Figure 1 with a resolution of 600 dpi. Moreover, we have extended Supplementary Figures 1c and 3c longitudinally as much as we can. 

 

To Reviewer #2

We would like to sincerely thank you very much for the time and effort you have dedicated to providing insightful feedback on ways to strengthen our manuscript entitled “Causal effect of atrial fibrillation/flutter on chronic kidney disease: A two-sample (bidirectional) Mendelian randomization study” under consideration for publication in PLOS ONE.

First, we would like to sincerely apologize for our major mistake and to correct it. 

Among 178 SNPs used as IVs for the change of eGFR in the original version of Supplementary Table 2, a total of 34 SNPs were not associated with eGFR at a genome-wide significance threshold (P = 5E-08) and they should have been excluded from the subsequent analyses (for example, rs10197255, P = 4.7E-07 in the original version of Supplementary Table 2).

To satisfy IV assumption 1 (the IVs are associated with the exposure), we should have selected SNPs as IVs that were associated with the exposure trait with P < 5.0×10-8 from the exposure GWAS summary data.

This error was due to the default setting of the clump_data function in the TwoSampleMR package (https://mrcieu.github.io/TwoSampleMR/reference/clump_data.html).

We originally selected the SNPs used as IVs for the change of eGFR from Supplementary Table 4 (ST4) in the GWAS study by CKDgen (Please refer to Nat Genet. 2019;51: 957-972, https://www.ncbi.nlm.nih.gov/pmc/articles/PMC6698888/).

Some of the 308 SNPs listed in ST4 were not associated with eGFR at a genome-wide significance threshold (that is, P > 5.0×10-8) based on populations of European ancestry.

And the default setting of the clump_data() function was as follows: 

clump_data(dat, 

 clump_kb = 10000,

 clump_r2 = 0.001,

 clump_p1 = 1,

 clump_p2 = 1,

 pop = "EUR"

)

Therefore, the 34 wrong SNPs with P > 5.0×10-8 were not excluded from our study by LD clumping. 

(Please refer to the paragraph of "Estimating the reverse causal effect of the change in eGFR on the risk of AF/F" in the original version of the Result session.)

Now, we have changed the setting of the clump_data() function as follows:

clump_data(dat,

 clump_kb = 10000,

 clump_r2 = 0.001,

 clump_p1 = 5E-08,

 clump_p2 = 1,

 pop = "EUR"

) 

And we have excluded a total of 158 SNPs by LD clumping including the 34 wrong SNPs.

(Please refer to the paragraph of "Estimating the reverse causal effect of the change in eGFR on the risk of AF/F" in the revised version of the Result session.)

Although nine SNPs have not been identified for the outcome GWAS datasets for AF/F, three SNPs were detected as proxy SNPs.

Then, we have used a total of 144 SNPs for estimating the causal effect of the change in eGFR on the risk of AF/F. (144 SNPs = 178 SNPs - 34 wrong SNPs.)

The 144 SNPs have been listed correctly in the revised version of Supplementary Table 2. 

Fortunately, the result of causality has been unchanged (the change in eGFR was not significantly associated with the risk of AF/F) though odds ratio, 95% CI, and P-value have been slightly changed.

Incorrect: OR, 0.997; 95% CI, 0.981-1.012; P = 0.67.

Correct: OR, 0.996; 95% CI, 0.980-1.013; P = 0.66. 

(Please see lines 246-247 in the revised manuscript.)

Incorrect: OR, 1.00081; 95% CI, 0.987-1.015; P = 0.91.

Correct: OR, 0.99967; 95% CI, 0.985-1.015; P = 0.97.

(Please see lines 251-252 in the revised manuscript.)

Please also refer to Table 2 in the revised manuscript.

Second, we would like to correct one more mistake.

IV assumption 1 was not "the IVs are associated with the outcome" but "the IVs are associated with the exposure". 

(Please see line 105 in the revised manuscript.)

Our responses to the Reviewer’s comments

Major points:

#1. Data source: CKDGen GWAS on eGFR

In this large meta-analysis, contributing studies derived residuals for log(eGFR) which then were used as outcome in the GWAS. It is incorrect to assume that this trait has SD of 1. This assumption was already incorrectly made by another study on which the Authors rely. Please revise.

We would like to apologize for our mistake and thank you very much for pointing it out.

As described in the CKDgen GWAS study that we referred to for the eGFR dataset, "exp(β) can be interpreted as the OR for the disease per unit change in log(eGFR)". 

(Please see the Methods section, the paragraph entitled "Genetic risk score analysis in the UK Biobank dataset" in Nat Genet. 2019;51: 957-972,

https://www.ncbi.nlm.nih.gov/pmc/articles/PMC6698888/).

Therefore, we corrected the effect size of the eGFR change on the AF/F risk as follows.

Incorrect: OR of AF/F per 1-SD higher log[eGFR] 

Correct: OR of AF/F per unit change in log[eGFR] 

Please see line 247 in the revised manuscript.

#2. Palindromic SNPs

Authors define palindromic SNPs with minor allele frequency (MAF) >0.42 and excluded thus some of the otherwise eligible SNPs. However, the Authors overlooked that a palindromic SNPs is also one with an effect allele frequency (EAF) – as presented in Supplementary Tables 1 and 2 - of, for example, 0.55. Since there are several SNPs with an EAF in the range of 0.5 to 0.58, exclusion of SNPs is incomplete. The analysis should be thus revised.

Please accept our apologies. We did not explain it clearly.

Supplementary Tables 1 and 2 originally included palindromic SNPs that were excluded from the subsequent analysis. 

We have marked all the excluded palindromic SNPs "TRUE" in the "palindromic ambiguous" columns in the revised version of Supplementary Tables 1 and 2.

Please also see lines 173-174, and 242 in the revised manuscript.

#3. Estimation of R² and F-statistics

Especially regarding R², I am not familiar with this presentation of formula. The reference provided by the Authors is also only an application that in turn point to a reference of Palmer et al 2012. Tracing that paper, however, does not provide the explanation on the formula. While the formula might be correct (I know presentations that look similar to a certain degree), it would be good to see who this formula proposed and how it was derived.

The main reason for my search was that a proper R² incorporates the variance of the outcome. In case such as in a 2-sample MR study the variance is often not available why usually some simplifying assumption such as SD=1 is applied. Since this is mostly likely not true for eGFR (see my first comment) and since I cannot deduce any other approximation of the variance of the outcome in the presented formula, I am wondering how reliable the estimates of R² and thus of the F-statistics are.

As – in the worst case – the strengths of instruments could be overestimated, this aspect needs clarification and discussion.

We sincerely apologize that our reference was inappropriate.

As Reviewer pointed out, the variance of the sex- and age-adjusted log(eGFR) residuals was assumed to be 0.016 (not = 1) as described in the CKDgen GWAS study (Nat Genet. 2019;51: 957-972, https://www.ncbi.nlm.nih.gov/pmc/articles/PMC6698888/).

However, we are sorry but we were not sure of the variance of the risk of AF/F.

Therefore, we could not use the formula (1) but used the formula (2) to estimate R² as follows:

(1); R² = (Beta)²×2×(EAF)×(1-EAF)/variance. 

(2); R² = 2×(Beta)²×EAF×(1-EAF)/[2×(Beta)²×EAF×(1-EAF) + 2×(SE)²×N×EAF×(1-EAF)].

The formula (2) that we used to estimate R² for each SNP was originally derived from the study by Sim H et al. (Please refer to S1 Text in PLoS One. 2015; 10(4): e0120758. https://www.ncbi.nlm.nih.gov/pmc/articles/PMC4405269/). 

For example, using the formula (2), R² of rs10159261 (the first row and the "O" column in the revised version of Supplementary Table 2) for the exposure (the change of eGFR) (Beta = -0.003774, SE = 0.00038, EAF = 0.31, N = 525153) was calculated as follows.

R² = 2×(-0.003774)²×0.31×(1-0.31)/[2×(-0.0038)²×0.31×(1-0.31) + 2×(0.00038)²×525153×0.31×(1-0.31)] = 0.0001877885.

R² of the other SNPs for the exposure datasets were also calculated using the formula (2) in the revised version of Supplementary Table 1 and 2. 

On the other hand, we performed the MR Steiger filtering method using steiger_filtering function in the TwoSampleMR package

(https://mrcieu.github.io/TwoSampleMR/reference/steiger_filtering.html) to infer the causal direction of each SNP on the hypothesized exposure and outcome.

(Please see lines 127-133, lines 174-176, lines 207-208, and lines 242-244 in the revised manuscript.)

Please also refer to point #1 for Reviewer #1.

The steiger_filtering function automatically calculated R² for the exposure and outcome dataset. 

When the SNPs for the change of eGFR were used as the exposure, the results of steiger_filtering function were as follows (e.g. the first 10 SNPs were shown): 

 SNP rsq.exposure

1 rs10159261 1.878319e-04

2 rs10254101 5.305158e-04

3 rs1028455 5.570221e-05

4 rs1047891 5.453993e-04

5 rs1055256 9.377984e-05

6 rs10821905 1.343213e-04

7 rs10846157 1.206184e-04

8 rs10857788 1.129092e-04 

9 rs10865189 9.680874e-05

10 rs11062167 2.595056e-04

Although we are not sure how the steiger_filtering function estimated R² values, the R² values estimated by the steiger_filtering function were almost the same as those by the formula (2) as shown in the revised version of Supplementary Table 2 (for example, R² of rs10159261 by the formula (2) and by steiger_filtering function were 0.0001877885 and 1.878319e-04, respectively, as shown above).

Therefore, we suppose that our R² estimation by the formula (2) and then F-statistics did not differ from the true values very much, and that our MR study was unlikely to suffer from weak instrument bias.

Please see lines 238-239 and 317-318 in the revised manuscript.

#4. Post-hoc power calculations

As discussed by many authors/statisticians (e.g. Dziak et al 2020, PMID: 32523323), post-hoc power calculations as presented in this paper are not valid. This part needs to be deleted and the topic differently approached.

We quite agree with Reviewer on this point. 

As Reviewer pointed out, we should have run the power calculations prior to conducting the analyses. 

We have deleted them totally (including Table 3) from our study in the revised manuscript. 

We suppose that our MR study was unlikely to suffer from weak instrument bias although our sample size of the AF/F dataset was relatively small.

In the MR study estimating the causal effect of the change in eGFR on the risk of AF/F, we suppose that we could not detect the causal effect, if any existed, because the scale of the OR (0.996) was very small.

We have reflected these comments to the revised manuscript as follows. 

On the other hand, we could not detect, if any existed, the causal effect of the change in eGFR on the risk of AF/F probably because the scale of the OR was very small. The F-statistic of every SNP > 28 indicated that there was no weak instrument bias (S2 Table).

(Please see lines 315-318 in the revised manuscript.)

Minor points:

#1. Since both directions are assessed and the standard phrasing is that of a “bidirectional” MR study it should be stated in the title as well as in the introduction.

We agree with Reviewer on this point.

We have made it clear that our MR study is bidirectional thorough the manuscript including the title. 

(Please see lines 3, 20, 50, and 58 in the revised manuscript.)

#2. Please provide phenotype definitions as used in the underlying studies.

Thank you very much for your suggestion.

We have added phenotype definition that was described in the CKDgen GWAS (Nat Genet. 2019;51: 957-972, https://www.ncbi.nlm.nih.gov/pmc/articles/PMC6698888/) as follows:

Serum creatinine assays were described in the GWAS study. GFR was estimated using the Chronic Kidney Disease Epidemiology Collaboration equation on adults (> 18 years of age) and the Schwartz formula on individuals who were 18 years old or younger, respectively.

(Please see lines 72-75 in the revised manuscript.)

The definition of CKD (defined as eGFR < 60 ml/min/1.73m2) and AF/F (ICD-10 code I48) was originally described in the manuscript.

(Please see lines 75-76 and 92-93 in the revised manuscript.)

#3. Please add further information on SNPs in Supplementary Tables 1 and 2, e.g., marking palindromic SNPs, outliers and otherwise noticeable SNPs as well as adding Phenoscanner information.

Thank you very much for your suggestion.

We added the further information about the SNPs as well as “Steiger direction” in the revised version of Supplementary Table 1 and 2 as follows: 

“Palindromic ambiguous”, the SNP was excluded if “TRUE.” (Please also see lines 173-174, and 242 in the revised manuscript);

“MR-PRESSO outlier”, outlier SNP was marked “Yes.”; 

“Sample size”; 

“Possible pleiotropic effects of the SNP for the exposure on other diseases and traits”;

“Steiger direction”, if “TRUE”, the SNP used as IV for the exposure was more predictive of the exposure than the outcome.

(Please also refer to the point #1 for Reviewer #1 as for “Steiger direction”).

#4. Please do Phenoscanner look up for all SNPs. Please be cautious in your wording of “unrelated traits”. Sometimes relation may not be so obvious.

We would like to appreciate this important suggestion.

We rephrased “unrelated to” as follows:

Incorrect: SNPs associated with P < 5.0×10-8 with pleiotropic effects on additional traits unrelated to AF/F.

Correct: SNPs associated with P < 5.0×10-8 with possible pleiotropic effects on other diseases and traits.

(Please see lines 158, 220-221, and 253 in the revised manuscript.)

Then, we have performed PhenoScanner serrch for 144 instrumental SNPs for the change in eGFR as well as re-performed for 20 instrumental SNPs for the risk of AF/F.

(Please see lines 157-159 in the revised manuscript.)

We found four SNPs for the AF/F dataset and 89 SNPs for eGFR dataset. 

The results were shown in the revised version of Supplementary Table 1 and 2.

When we excluded the four SNPs from the IVW method to estimate the causal effect of the risk of AF/F on CKD risk, we obtained a comparable result to that of the original IVW method (OR of CKD per log OR of AF/F, 25.3 vs 33.2; 95% CI, 3.51-183.0 vs 2.07-532.2; P = 0.001 vs 0.013).

(Please see lines 220-226 in the revised manuscript.) 

Please also see lines 301-304 in the revised manuscript. 

Similarly, when we excluded the 89 SNPs from the IVW method to estimate the causal effect of the change in eGFR on the AF/F risk, we obtained a comparable result to that of the original IVW method (OR of CKD per log OR of AF/F, 0.996 vs 1.005; 95% CI, 0.980-1.013 vs 0.980-1,031; P = 0.66 vs 0.69).

(Please see lines 252-256 in the revised manuscript.) 

#5. In presentation of results on the evaluation of the causal relationship of AF/F on eGFR, the Authors state that they used a multiplicative random effects model because of observed heterogeneity. Since the results on IVW estimate is presented above, this statement needs clarification if the presented result is already from multiplicative random effects model, also in contrast to all other presented results from IVW analysis. Was this the only instance?

We would like to apologize for any confusion we have caused and to appreciate the suggestion.

In a similar way to the MR study conducted by Dr. Jie V Zhao et al [29], we re-analyzed the IVW methods thorough our study using a multiplicative random-effects model when Cochran’s Q statistic was significant (P < 0.05). Otherwise, a fixed-effects model was used. 

We have stated it clearly in the Methods and Result section of the revised manuscript (Please see lines 139-141, 177, 184-187, 209, 224-225, 226-229, 245, 251, 255, and 256-258 in the revised manuscript.)

(Reference)

29. Zhao JV, Schooling CM. Effect of linoleic acid on ischemic heart disease and its risk factors: a Mendelian randomization study. BMC Med. 2019;17: 61.

Accordingly, when we excluded an outlier SNP (rs796427) by MR-PRESSO from our MR analysis estimating the causal effect of the risk of AF/F on the change in eGFR, Cochran’s Q statistic for the IVW method indicated low heterogeneity (24.3, P = 0.11).

Therefore, we re-analyzed the IVW method using a fixed-effects model.

As a result, Beta (ORs) remained unchanged, but SE got smaller (from 0.0455 to 0.038) with significant significance (P = from 0.036 to 0.012 < 0.017).

We corrected it in the Result session (lines 183-187, and 203-205), the Discussion session (line 333), and in Table 1 of the revised manuscript.

Similarly, in the analysis estimating the causal effect of the risk of AF/F on CKD risk, Cochran’s Q statistic for the IVW method indicated low heterogeneity (23.0, P = 0.19) as shown originally in Table 1.

Therefore, we re-analyzed the IVW method using a fixed-effects model.

As a result, Beta (ORs) remained unchanged, but SE got smaller (95% CI got narrower) with significant significance.

Then we corrected the result as follows.

Incorrect: OR of CKD per log OR of AF/F, 25.3; 95% CI, 2.71-236.9; P = 0.0046.

Correct: OR of CKD per log OR of AF/F, 25.3; 95% CI, 3.51-183.0; P = 0.001.

Incorrect: OR of CKD was 9.39 per doubling OR of AF/F (95% CI, 1.99-44.2).

Correct: OR of CKD was 9.39 per doubling OR of AF/F (95% CI, 2.39-37.0).

(Please see lines 27 and 209-212 in the revised manuscript.)

Please also refer to Table 1 in the revised manuscript.

#6. The Authors state in the Discussion that estimates may be too large and CIs were too wide and these present a major limitation of the study. However, the reported results present maybe a consequence of some limitations but do not present a limitation themselves. Please revise.

We quite agree with Reviewer on this point.

The relatively small sample size of the AF/F dataset was one of our major limitations.

We have rephrased this part as follows:

Our MR estimate scale of OR of CKD per doubling OR of AF/F may be too large and the corresponding 95% CI width was very wide (OR, 9.39; 95%CI, 1.99-44.2). One possible reason for this observation was the relatively small sample size of the AF/F dataset that was one of the major limitations of this study, as we could not use the largest GWAS meta-analysis due to substantial sample overlap.

(Please see lines 307-311 in the revised manuscript.)

Please also see lines 270 and 285-286 in the revised manuscript.

---

## [Decision Letter · Decision Letter 1]

1 Sep 2021

PONE-D-21-11069R1

Causal effect of atrial fibrillation/flutter on chronic kidney disease: A bidirectional two-sample Mendelian randomization study

PLOS ONE

Dear Dr. Yoshikawa,

Thank you for submitting your manuscript to PLOS ONE. After careful consideration, we feel that it has merit but does not fully meet PLOS ONE’s publication criteria as it currently stands. Therefore, we invite you to submit a revised version of the manuscript that addresses the points raised during the review process. Please refer to the comments raised by the reviewers and revise accordingly.

We look forward to receiving your revised manuscript.

Kind regards,

Jie V Zhao

Academic Editor

PLOS ONE

Journal Requirements:

Reviewers' comments:

Reviewer's Responses to Questions

**Comments to the Author**

1. If the authors have adequately addressed your comments raised in a previous round of review and you feel that this manuscript is now acceptable for publication, you may indicate that here to bypass the “Comments to the Author” section, enter your conflict of interest statement in the “Confidential to Editor” section, and submit your "Accept" recommendation.

Reviewer #1: (No Response)

Reviewer #2: (No Response)

2. Is the manuscript technically sound, and do the data support the conclusions?

Reviewer #1: Yes

Reviewer #2: Yes

3. Has the statistical analysis been performed appropriately and rigorously? 

Reviewer #1: Yes

Reviewer #2: Yes

4. Have the authors made all data underlying the findings in their manuscript fully available?

Reviewer #1: Yes

Reviewer #2: Yes

5. Is the manuscript presented in an intelligible fashion and written in standard English?

Reviewer #1: Yes

Reviewer #2: Yes

6. Review Comments to the Author

Reviewer #1: The authors have adequately addressed my comments.

Figure 1 is still hard to read. Please revise.

Reviewer #2: Causal effect of atrial fibrillation/flutter on chronic kidney disease: A two-sample Mendelian randomization study

By Masahiro Yoshikawa, Kensuke Asaba and Tomohiro Nakayama

Reviewer Comments

I thank the Authors for their thorough job addressing all my comments. Explanations were provided, corrections were made. Only one last point I still do not understand and concerns my previous point on palindromic SNPs. While Authors’ explanation are understandable I still have trouble:

In Supplementary Table 1, one SNP (rs7853195) is marked as ‘palindromic ambiguous’ and is mentioned to be excluded from the analysis in the Results (line 173). This can be retraced by the fact that the allele frequency of the effect allele is 0.42 in both GWAS of exposure and outcome. However, there is another SNP (rs4245712) in that table that is not marked as ‘palindromic ambiguous’ but has allele frequency of the effect allele of 0.55 (exposure) and 0.56 (outcome). Since MAF is >0.42 (definition of palindromic SNPs, lines 119-120) I do not understand why this SNP was not excluded. In Supplementary Table 2, there are also such instances (e.g., rs1055256). What do I miss? Did the Authors used a different source than the GWAS to define palindromic SNPs or incorporated some additional criteria?

Otherwise, I am quite satisfied with the presentation of the project.

7. PLOS authors have the option to publish the peer review history of their article (what does this mean?). If published, this will include your full peer review and any attached files.

Reviewer #1: No

Reviewer #2: No

---

## [Author Response · Author response to Decision Letter 1]

8 Sep 2021

Dear Dr. Jie V Zhao, 

We would like to express our sincere gratitude and appreciation for the opportunity to submit a revision of our manuscript entitled "Causal effect of atrial fibrillation/flutter on chronic kidney disease: A bidirectional two-sample Mendelian randomization study".

Journal Requirements: Please review your reference list to ensure that it is complete and correct. If you have cited papers that have been retracted, please include the rationale for doing so in the manuscript text, or remove these references and replace them with relevant current references. Any changes to the reference list should be mentioned in the rebuttal letter that accompanies your revised manuscript. If you need to cite a retracted article, indicate the article’s retracted status in the References list and also include a citation and full reference for the retraction notice.

Reference 18 is not "Forthcoming" but "Online ahead of print" and we revised it (please see line 397 in the newest version of the manuscript).

Now we have confirmed that our reference list is correct without retracted papers.

And our responses to each Reviewer’s comment are described below in a point-to-point manner. 

We thank you very much for handling this paper and look forward to hearing from you again.

Sincerely, 

Masahiro Yoshikawa, M.D., Ph.D. 

Corresponding author.

Address; Division of Laboratory Medicine, Department of Pathology and Microbiology, Nihon University School of Medicine, Oyaguchi-kamicho 30-1, Itabashi, Tokyo, Japan. 

Phone; 81-3-3972-8111

e-mail: myosh-tky@umin.ac.jp

 

To Reviewer #1

Our responses to the Reviewer’s comments

The authors have adequately addressed my comments.

We would like to express our sincere gratitude and appreciation for your insightful suggestions.

We have learned a lot, and we believe that our manuscript has been significantly improved thanks to Reviewer #1.

Figure 1 is still hard to read. Please revise.

Could you please click "Click here to access/download;Figure;Fig1.tif " on the top right of Revised Figure 1 page to download TIF file with a resolution of 600 dpi? 

 

To Reviewer #2

Our responses to the Reviewer’s comments

I thank the Authors for their thorough job addressing all my comments. Explanations were provided, corrections were made.

We would like to express our sincere gratitude and appreciation for pointing out our mistakes and faults. We also thank you very much for insightful suggestions.

We believe that our manuscript has been significantly improved thanks to Reviewer #2.

Only one last point I still do not understand and concerns my previous point on palindromic SNPs. While Authors’ explanation are understandable I still have trouble: In Supplementary Table 1, one SNP (rs7853195) is marked as ‘palindromic ambiguous’ and is mentioned to be excluded from the analysis in the Results (line 173). This can be retraced by the fact that the allele frequency of the effect allele is 0.42 in both GWAS of exposure and outcome. However, there is another SNP (rs4245712) in that table that is not marked as ‘palindromic ambiguous’ but has allele frequency of the effect allele of 0.55 (exposure) and 0.56 (outcome). Since MAF is >0.42 (definition of palindromic SNPs, lines 119-120) I do not understand why this SNP was not excluded. In Supplementary Table 2, there are also such instances (e.g., rs1055256). What do I miss? Did the Authors used a different source than the GWAS to define palindromic SNPs or incorporated some additional criteria? 

Thank you very much for your careful reading and suggestion, and we sincerely apologize that our tables were still hard to read.

In the same way as MR-Base platform does, we have marked the palindromic SNPs "TRUE" in the "palindromic" column and the excluded SNPs "TRUE" in the "ambiguous" column in the newest version of Supplementary Tables 1 and 2. Please also see lines 174-175 and 242-243 in the newest manuscript.

For example, rs4245712 in Supplementary Table 1 has effect allele frecency (EAF) of 0.55 for exposure and 0.56 for outcome (i.e. minor allele frequency (MAF) > 0.42), but it is not a palindromic SNP ("palindromic" is "FALSE"). Therefore, it was not excluded.

In Supplementary Table 2, EAF of rs4820324 for exposure is, more precisely, 0.5801 (Please see row 308 and column N in ST4 in the GWAS study by CKDgen, Nat Genet. 2019;51: 957-972, https://www.ncbi.nlm.nih.gov/pmc/articles/PMC6698888/). And EAF of rs4820324 for outcome is 0.580514. Both are slightly > 0.58 (i.e. MAF < 0.42), and therefore it was not excluded from our analyses.

On the other hand, rs7853195 in Supplementary Table 1 in our manuscript (EAF is 0.421733 for exposure, slightly > 0.42) was excluded properly.

Otherwise, I am quite satisfied with the presentation of the project.

We really appreciate your help in improving our manuscript.

---

## [Decision Letter · Decision Letter 2]

8 Oct 2021

PONE-D-21-11069R2Causal effect of atrial fibrillation/flutter on chronic kidney disease: A bidirectional two-sample Mendelian randomization studyPLOS ONE

Dear Dr. Yoshikawa,

Thank you for submitting your manuscript to PLOS ONE. After careful consideration, we feel that it has merit but does not fully meet PLOS ONE’s publication criteria as it currently stands. Therefore, we invite you to submit a revised version of the manuscript that addresses the points raised during the review process.

We look forward to receiving your revised manuscript.

Kind regards,

Jie V Zhao

Academic Editor

PLOS ONE

Journal Requirements:

Reviewers' comments:

Reviewer's Responses to Questions

**Comments to the Author**

1. If the authors have adequately addressed your comments raised in a previous round of review and you feel that this manuscript is now acceptable for publication, you may indicate that here to bypass the “Comments to the Author” section, enter your conflict of interest statement in the “Confidential to Editor” section, and submit your "Accept" recommendation.

Reviewer #1: (No Response)

Reviewer #2: All comments have been addressed

2. Is the manuscript technically sound, and do the data support the conclusions?

Reviewer #1: Yes

Reviewer #2: (No Response)

3. Has the statistical analysis been performed appropriately and rigorously? 

Reviewer #1: Yes

Reviewer #2: (No Response)

4. Have the authors made all data underlying the findings in their manuscript fully available?

Reviewer #1: Yes

Reviewer #2: (No Response)

5. Is the manuscript presented in an intelligible fashion and written in standard English?

Reviewer #1: Yes

Reviewer #2: (No Response)

6. Review Comments to the Author

Reviewer #1: Figure 1 is still hard to read. I don't think the author's response that "Could you please click "Click here to access/download;Figure;Fig1.tif " on the top right of Revised Figure 1 page to download TIF file with a resolution of 600 dpi?" is acceptable. If one chooses to print out a hard copy to read, then where to click?

Reviewer #2: (No Response)

7. PLOS authors have the option to publish the peer review history of their article (what does this mean?). If published, this will include your full peer review and any attached files.

Reviewer #1: No

Reviewer #2: No

---

## [Author Response · Author response to Decision Letter 2]

9 Nov 2021

To Reviewer #1

Figure 1 is still hard to read. I don't think the author's response that "Could you please click "Click here to access/download;Figure;Fig1.tif " on the top right of Revised Figure 1 page to download TIF file with a resolution of 600 dpi?" is acceptable. If one chooses to print out a hard copy to read, then where to click?

We would like to apologize for any inconvenience.

We have separated original Figure 1(a-c) into revised Figures 1-3 to increase the resolution.

We sincerely hope that new Figures could meet with your criteria.

---

## [Editor Report · Decision Letter 3]

23 Nov 2021

Causal effect of atrial fibrillation/flutter on chronic kidney disease: A bidirectional two-sample Mendelian randomization study

PONE-D-21-11069R3

Dear Dr. Yoshikawa,

We’re pleased to inform you that your manuscript has been judged scientifically suitable for publication and will be formally accepted for publication once it meets all outstanding technical requirements.

Kind regards,

Jie V Zhao

Academic Editor

PLOS ONE
---

## [Editor Report · Acceptance letter]

29 Nov 2021

PONE-D-21-11069R3 

Causal effect of atrial fibrillation/flutter on chronic kidney disease: A bidirectional two-sample Mendelian randomization study 

Dear Dr. Yoshikawa:

I'm pleased to inform you that your manuscript has been deemed suitable for publication in PLOS ONE. Congratulations! Your manuscript is now with our production department. 

Kind regards, 

on behalf of

Dr. Jie V Zhao 

Academic Editor

PLOS ONE